# PARP Inhibitors in Breast and Ovarian Cancer

**DOI:** 10.3390/cancers15082357

**Published:** 2023-04-18

**Authors:** Samuel S. Y. Wang, Yeo Ee Jie, Sim Wey Cheng, Goh Liuh Ling, Heong Valerie Yue Ming

**Affiliations:** 1Medical Oncology, Tan Tock Seng Hospital, Singapore 308433, Singapore; 2Molecular Diagnostic Laboratory, Tan Tock Seng Hospital, Singapore 308433, Singapore

**Keywords:** PARP inhibitor, breast cancer, ovarian cancer, BRCA1, BRACA2

## Abstract

**Simple Summary:**

Targeted therapy has revolutionised oncological treatments in cancer with great efficacy and a more tolerable side effect profile as compared to standard chemotherapy. Poly (ADP-ribose) polymerase (PARP) inhibitors has demonstrated in their effective management of BRCA1/BRCA2 mutant cancers, most notably in breast and ovarian cancers. However, cancer cells have the innate ability to mutate to create resistance against therapy. In this review, we will be discussing on the efficacy of PARP inhibitors, mechanisms of mutation, as well as ways to overcome resistance.

**Abstract:**

Poly (ADP-ribose) polymerase (PARP) inhibitors are one of the most successful examples of clinical translation of targeted therapies in medical oncology, and this has been demonstrated by their effective management of BRCA1/BRCA2 mutant cancers, most notably in breast and ovarian cancers. PARP inhibitors target DNA repair pathways that BRCA1/2-mutant tumours are dependent upon. Inhibition of the key components of these pathways leads to DNA damage triggering subsequent critical levels of genomic instability, mitotic catastrophe and cell death. This ultimately results in a synthetic lethal relationship between BRCA1/2 and PARP, which underpins the effectiveness of PARP inhibitors. Despite the early and dramatic response seen with PARP inhibitors, patients receiving them often develop treatment resistance. To date, data from both clinical and preclinical studies have highlighted multiple resistance mechanisms to PARP inhibitors, and only by understanding these mechanisms are we able to overcome the challenges. The focus of this review is to summarise the underlying mechanisms underpinning treatment resistance to PARP inhibitors and to aid both clinicians and scientists to develop better clinically applicable assays to better select patients who would derive the greatest benefit as well as develop new novel/combination treatment strategies to overcome these mechanisms of resistance. With a better understanding of PARP inhibitor resistance mechanisms, we would not only be able to identify a subset of patients who are unlikely to benefit from therapy but also to sequence our treatment paradigm to avoid and overcome these resistance mechanisms.

## 1. Introduction

The hallmarks of cancer are comprised of sustaining proliferative signalling, evading growth suppressors, resisting cell death, enabling replicative immortality, inducing angiogenesis, and activating invasion and metastasis [1]. However, underpinning these hallmarks is genome instability, which generates the genetic diversity that expedites and fosters multiple hallmark functions [1]. Genomic instability often results from altered DNA repair capabilities resulting in different cancer types. The poly (ADP-ribose) polymerase (PARP) enzymes are critical enzymes involved in DNA repair and several cellular processes, including DNA replication, chromatin remodelling and apoptosis [2]. PARP1 and PARP2 have been extensively characterised for their involvement in the DNA repair processes. The PARP1 enzyme is involved in base excision repair (BER) in response to single-strand DNA breaks (SSBs) [3]. PARP1 has also been reported to play a role in alternative DNA repair mechanisms, including the nucleotide excision repair (NER) [4]. However, PARP1 inhibition alone is not lethal as the DNA damage caused by PARP1 inhibition can be repaired via other repair pathways, mainly homologous recombination (HR). The breast-cancer-associated genes 1 and 2 (*BRCA1* and *BRCA2*) mediate HR, a DNA repair mechanism for double-strand DNA breaks [5,6], and the loss of either one of these genes coupled with the inactivation of PARP1 results in synthetic lethality and cell death as cells with mutations in *BRCA1* or *BRCA2* are unable to effectively perform HR [7,8]. The initial studies observed that decreasing PARP1 levels using RNA interference resulted in a significant reduction in cell survival selectively in BRCA-deficient cells [9]. Furthermore, it was shown that cancer cell lines lacking *BRCA1/2* were sensitive to inhibitors of PARP1, whereas cells with functional *BRCA1/2* were not [8,9]. This was followed subsequently by several studies that showed that the loss of other tumour suppressor DNA repair proteins involved in HR repair may also cause sensitization to PARPi [10,11].

A subsequent first-in-human clinical evaluation of PARPi was performed in a phase 1 clinical trial of PARPi olaparib; pharmacodynamic and pharmacokinetic evaluations were performed to optimize drug doses and dose-limiting toxicities were recorded. Sixty patients with different cancer types were enrolled and treated. Of these, 23 patients were confirmed germline *BRCA1/2* mutation carriers (g*BRCA1/2*); however, two of these patients could not be evaluated with regard to antitumour response due to toxicity and stopping the drug after less than 4 weeks, while another two had tumours not typically associated with BRCA carrier status: one with small-cell lung cancer and one with vaginal adenocarcinoma. The remaining 19 gBRCA carriers had ovarian, breast or prostate cancers; 63% (12 of the 19) derived clinical benefit from olaparib treatment, with radiologic or tumour-marker responses or meaningful disease stabilization (stable disease for a period of 4 months or more). A total of 47.4% (9 out of 19) of BRCA carriers had a response according to RECIST criteria. Strikingly, in patients without known *BRCA1/2* mutations, no objective antitumour responses were observed. More importantly, patients harbouring g*BRCA1/2* mutations did not experience enhanced toxicities, supporting the hypothesis of a selective cancer vulnerability [12]. The discovery of synthetic lethality due to the combined loss of PARP1/inhibition of PARP1 with a PARPi and *BRCA1/2* defect has revolutionized the treatment of DNA-repair-deficient cancers mainly in ovarian, breast, prostate and pancreatic cancers and has resulted in the development of a number of PARP inhibitors namely olaparib, niraparib, rucaparib, talazoparib and veliparib across different tumour types [12,13].

## 2. PARP Inhibitors across Different Tumour Types

Following the first phase 1 study of PARPi therapy [12], cancer therapy has pivoted towards precision medicine and biomarker-selected therapies in an effort to limit toxicity and allow better selection of patients. This resulted in an increasing number of PARPi clinical trials across multiple settings in different tumour types. The benefit for monotherapy PARPi was observed to be consistent in germline *BRCA1/2* mutation carriers across different tumour types regardless of the number of lines of therapy. This was particularly evident from a multicentre phase II trial of g*BRCA1/2* carriers with heavily pretreated breast, ovarian, prostate or pancreatic cancer who had progressed through multiple lines of chemotherapy and were treated with PARPi, olaparib. Overall response rates were 26.2%, with the highest responses being in prostate cancer (50%) and ovarian cancer (31.1%), far outweighing any response to conventional chemotherapy in the same setting [14].

### 2.1. PARP Inhibitor Therapy in Breast Cancer

In 2017, a phase III study in locally advanced or metastatic HER2 negative, g*BRCA1/2* mutation-associated breast cancer patients (OlympiAD) were randomised to monotherapy olaparib versus physician’s choice chemotherapy (capecitabine, gemcitabine, eribulin or vinorelbine). The trial demonstrated improved progression-free survival (PFS) in the olaparib arm compared with patients who received the physician’s choice, 19.3 months with olaparib versus 17.1 months with the physician’s choice (HR 0.90, 95% CI 0.66–1.23; *p* = 0.513). Reassuringly, olaparib was better tolerated than chemotherapy (grade ≥ 3 was 38.0% in the olaparib arm vs. 49.5% in the capecitabine, vinorelbine or eribulin arm), and had a tolerable safety profile with only a 4.7% discontinuation rate due to toxicity [15]. Interestingly, patients who received olaparib in an earlier line of treatment had a trend towards improved overall survival (OS) (22.6 versus 14.7 months; HR  =  0.51; 95% CI: 0.29–0.90) possibly related to potential resistance mechanisms. The superiority of PARPi in g*BRCA1/2* mutation-associated breast cancer was reaffirmed in another phase III study with talazoparib compared with a similar standard single agent chemotherapy. This trial showed a median PFS of 8.6 months vs. 5.6 months (hazard ratio for disease progression or death, 0.54; 95% confidence interval (CI), 0.41 to 0.71; *p* < 0.001) and objective response rate of 62.6% compared to 27.2%), and similar toxicity profiles, respectively [16]. Overall survival benefit was not demonstrated in either of these studies.

There have been several trials that have investigated these therapies in the earlier stages of disease. The BrighTNess Trial showed that adding a PARPi veliparib to carboplatin-containing neoadjuvant chemotherapy did not impact long-term outcomes in TNBC [17]. Conversely, another phase 3 randomised trial (OlympiA) demonstrated that by adding the PARPi olaparib in the adjuvant setting for patients with a high risk of relapse, there was an improved three-year invasive-free survival of 85.9% vs. 77.1% in the placebo group [18] (Table 1).

### 2.2. PARP Inhibitor Therapy in Ovarian Cancer

PARPi has had a role to play as a single agent maintenance therapy, with great responses for both *BRCA1/2* carriers and non-carriers. Niraparib is used for maintenance therapy irrespective of BRCA status, and its efficacy was demonstrated in the Prima trial with significantly longer progression-free survival compared to a placebo among patients with homologous recombination deficiency (21.9 month vs. 10.4 month, HR 0.4) [19]. Building upon the Prima trial, there was an interest in evaluating the effectiveness of niraparib in germline BRCA mutation for platinum-sensitive, recurrent ovarian cancer. For the germline BRCA cohort, the overall PFS was 21.0 months in the niraparib arm vs. 5.5 months in the placebo arm (HR, 0.27; 95% CI, 0.17 to 0.41), while for the non-germline BRCA cohort the PFS was 12.9 months in the niraparib arm vs. 3.8 months in the placebo arm (HR, 0.45; 95% CI, 0.34 to 0.61; *p* < 0.001) [26]. Further evaluation was made for non-germline *BRCA* patients with homologous recombination deficiency where PFS was 12.9 months for the niraparib arm vs. 3.8 months in the placebo arm (HR, 0.38; 95% CI, 0.24 to 0.59) [26].

The SOLO1 trial demonstrated olaparib to be an effective maintenance therapy treatment option for patients with newly diagnosed, advanced high-grade serous or endometrioid ovarian cancer with *BRCA1/2* mutation and had achieved complete or partial response after platinum-based chemotherapy, with an impressive median PFS of 50.6 months [20]. Building on the SOLO1 trial, the SOLO2 trial aimed to investigate olaparib’s effectiveness as a maintenance therapy in sensitive, relapsed high-grade serous ovarian cancer with *BRCA1/2* mutation; the results showed that olaparib was associated with a significantly longer PFS (19.1 months [95% CI 16.3–25.7]) than with a placebo (5.5 months [5.2–5.8]; hazard ratio (HR) 0.30 [95% CI 0.22–0.41], *p* < 0.0001) [24]. Other trials such as study 19 evaluated the effectiveness of olaparib in all comers in the maintenance setting in relapsed platinum sensitive ovarian cancer. Specifically, [19] focused on platinum-sensitive, relapsed, high-grade serous ovarian cancer, and the PFS was significantly longer with olaparib than with a placebo (median, 8.4 months vs. 4.8 months, respectively, from randomization on completion of chemotherapy; hazard ratio for progression or death, 0.35; 95% confidence interval (CI), 0.25 to 0.49; *p* < 0.001) [25].

PARPi has also been used in maintenance treatment for newly diagnosed ovarian cancer in a combination with other drugs in an attempt to achieve additional benefit. The PAOLA-1 study, with a median follow-up of 22.9 months demonstrated a median PFS of olaparib plus bevacizumab of 22.1 months vs. 16.6 for bevacizumab alone (HR 0.59, 95% CI, 0.49 to 0.72; *p* < 0.001) as a first-line maintenance therapy for patients with homologous recombination deficient newly diagnosed, advanced, high-grade ovarian cancer, with a response after first-line platinum–taxane chemotherapy plus bevacizumab [21]. ATHENA-MONO also explored rucaparib as monotherapy maintenance therapy for patients with stage III-IV high-grade ovarian cancer undergoing surgical cytoreduction (R0 resection) and responding to first-line platinum-doublet chemotherapy. The greatest benefit was seen for patients with homologous recombination deficiency, with a PFS of 28.7 months with rucaparib vs. 11.3 months with a placebo (*p* = 0.004; HR 0.47; 95% CI 0.31–0.72) [22].

The ARIEL 2 study evaluated the effectiveness of rucaparib in recurrent, platinum-sensitive, high-grade ovarian carcinoma with either BRCA1 and BRCA2 (BRCA) mutations or genomic loss of heterozygosity (LOH) scores. The largest benefit was seen in the BRCA mutant cohort with the median PFS of 12.8 months (95% CI 9.0–14.7) in the BRCA mutant subgroup. The other cohorts did see a benefit although the benefit was substantially smaller with PFS of 5.7 months (5.3–7.6) in the LOH high subgroup and 5.2 months (3.6–5.5) in the LOH low subgroup [27]. PFS was significantly longer in the BRCA mutant (hazard ratio 0.27, 95% CI 0.16–0.44, *p* < 0.0001) followed by the LOH high group (0.62, 0.42–0.90, *p* = 0.011) [27].

In the relapsed setting, SOLO3 and ARIEL 3 both showed benefit of PARPi when used in a treatment setting for relapsed ovarian cancer. SOLO3 demonstrated a superior objective response rate with olaparib when compared with chemo (72.2% vs. 51.4%; odds ratio (OR), 2.53 [95% CI, 1.40 to 4.58]; *p* = 0.002) in patients with BRCA-mutated platinum-resistant or partial platinum-sensitive relapsed ovarian cancers [23]. ARIEL3 looked at rucaparib treating recurrent ovarian cancers that had received at least two previous platinum chemotherapy treatments with a response, and the median PFS in patients with BRCA mutant carcinoma was 16.6 months (95% CI 13.4–22.9) vs. 5.4 months (95% CI 3.4–6.7) [28] (Table 1).

### 2.3. PARP Inhibitor Therapy in Other Tumour Types

Several clinical studies of PARPi therapy have been reported primarily for pancreatic and prostate cancer. Alterations in DNA repair genes including *BRCA1/2* are fairly common in metastatic prostate cancer with up to 20–25% of prostate cancers reported to harbour a defect in the DNA damage repair pathway [31], and retrospective analysis has reported a strong association between olaparib’s antitumour activity and mutations in different DNA repair genes [32]. A randomised, open label trial by de Bono et al. demonstrated that olaparib improved PFS (7.4 month vs. 3.6 month; HR for progression or death, 0.34; 95% CI, 0.25 to 0.47; *p* < 0.001) and OS (18.5 months vs. 15.1 months, respectively) when compared to enzalutamide or abiraterone, an androgen biosynthesis inhibitor that inhibits 17α-hydroxylase/C17,20-lyase (CYP17) [29]. Another double-blinded, randomised trial by Clarke et al. demonstrated that by adding olaparib to abiraterone, the radiological-PFS was improved to 13.8 months vs. 8.2 months compared to abiraterone alone, although at the expense of a greater side effect profile (54% vs. 28% rates of grade 3 or worse adverse events) [33] (Table 1).

In pancreatic cancer, a phase III study (POLO) of PARPi, olaparib, in the maintenance setting after response to platinum-based chemotherapy in patients with germline *BRCA1/2* mutation carriers demonstrated a significant improvement in PFS compared to a placebo (7.4 months vs. 3.8 months, respectively; HR 0.53; 95% CI, 0.35 to 0.82; *p* = 0.0038) with no significant difference in the change in global quality of life score [34,35]. This, however, did not affect overall survival [34,35] (Table 1).

Key points of the studies mentioned above are summarised in Table 1.

## 3. Challenges Facing PARP Inhibitors

Despite the unprecedented benefit of PARPi therapy (Table 1) in certain groups of patients, not all patients benefit equally from this therapy with up to 40% of *BRCA* mutant ovarian cancer patients failing to derive any benefit [36,37]. Moreover, continuous exposure to PARPi may result in cancer cells acquiring PARPi resistance [38,39]. Mutations that either cause functional recovery of BRCA or those that enable the homologous recombination (HR) pathway impact upon the effectiveness of PARPi therapy, as synthetic lethality hinges upon the activation of the non-homologous end-joining (NHEJ) pathway, which is responsible for error prone repairs in double-stranded breaks and leads to genomic instability and eventually apoptosis [40]. Several mechanisms underlie PARPi resistance including reversion mutations of HR genes, the restoration of HR via the inactivation of NHEJ proteins, the restoration of replication fork stability, decreased PARPi binding to the PARP1 protein and increased drug efflux activity [40].

Reversion mutations of HR genes may occur after prolonged exposure to PARPi and cisplatin; for example, the protein-truncating c.6174delT frameshift mutation of *BRCA2* in cancer cells was converted to restore the open reading frame (ORF), thereby restoring function to the BRCA2 gene despite not having full conversion to the wild type gene [41]. Interestingly, there have been no reports on reversion mutations in which all BRCA mutations have been completely eliminated, which suggests that minimal functioning BRCA2 genes are required to confer BRCA2 function.

Another subset of reversion mutations are secondary reversion mutations, where new mutations restore the ORF after the primary mutation causes a frameshift mutation. Secondary mutations have been noted in RAD51C and RAD51D, genes that are also implicated in HR, and these mutations occurred after rucaparib was administered and interestingly restored the RAD51 function and increased PARPi and platinum-based chemotherapy resistance [42].

Restoration of HR via the inactivation of NHEJ proteins is another method of PARPi resistance. For example, p53-binding protein 1 (53BP1) is an important NHEJ protein, and through the inactivation of 53BP1, it shifts the choice between NHEJ towards HR [43]. This loss of 53BP1 expression has been noted in BRCA1/2-mutated or triple-negative breast cancers and has been associated with a poorer survival rate [44]. It is therefore thought that measurement of 53BP1 might be a potentially useful biomarker to predict the BRCA mutant cancer response to PARPi agents. Other genes have also been implicated; for example, REV7/MAD2L2 is important for HR restoration and PARPi resistance in BRCA1-deficient cells [45]. REV7/MAD2L2 interacts with 53BP1 downstream and blocks DNA resection to promote NHEJ. Hence, when REV7/MAD2L2 is depleted, BRCA1-deficient cells acquire PARPi resistance. The cancer genome atlas database was utilised to calculate gene-specific survival probabilities for pancreatic cancer, ovarian cancer and breast cancer where PARPi are predominantly used for treatment. Higher REV7/MAD2L2 expression was associated with higher survival probabilities in pancreatic cancer, ovarian cancer and breast cancer, while low levels of REV7/MAD2L2 exhibited an unfavourable outcome [40].

PARPi resistance usually occurs with increased genomic instability, and in non-malignant cells, there lie multiple DNA fork protection mechanisms to promote genomic stability. Paradoxically, these mechanisms still remain active in malignant cells and interfere with the cytotoxic effect of anticancer therapeutics. Inactivation of the SNF2-family fork remodelers, including SMARCAL1, ZRANB3 and HLTF, which are critical for fork reversal, reduces replication stress and genomic instability, and provides PARPi resistance [46]. Survival probability panels in pancreatic cancer showed low expression of ZRANB3 and HLTF associated with poorer survival probabilities than those with high expression [40]. Similarly, in ovarian cancer, low expression of SMARCAL1 and PTIP exhibited an unfavourable outcome [40], while PTIP also showed similar survival probabilities in breast cancer [40]. FANCD2 overexpression conferred resistance to PARPi through fork stabilization, suggesting that FANCD2 can be used as a PARPi resistance biomarker [47].

PARPi might encounter resistance if their target proteins are either depleted or undergo mutations that prevent PARPi binding. In vitro evidence of PARP1 knockout cells showed high resistance to olaparib treatment [10], while genome-wide CRISPR–Cas9 mutagenesis screening identified several PARP1 mutations associated with PARPi resistance [48]. Interestingly, in ovarian cancer patients, the *PARP1* p.R591C mutation (c.1771C>T) showed de novo resistance to olaparib, showing clinical evidence of PARP mutations as another mechanism of PARPi resistance [48].

Another mechanism of PARPi resistance is to reduce the effective dose of PARPi available in the cancer cells. Cancer cells can do this by increasing the expression of ABCB1 genes, also referred to as multidrug resistance (MDR1) genes. This increases the availability of ATP-binding cassette transporters, such as the P-glycoprotein efflux pump, which helps remove the PARPi within the cancer cells [49].

## 4. Causes and Mechanisms of PARP Inhibitor Resistance

### 4.1. Restoration of Homologous Recombination

The overall efficacy of HR depends on the intact functions of *BRCA1*, *BRCA2* and *PALB2*. Mutations in *BRCA1*, *BRCA2* or *PALB2* work synergistically with PARPi to cause cancer cell death. HR-deficient cells almost always become resistant by developing reversion mutations that either restore or bypass HR [41,50]. Reverse mutations in *BRCA1/2* or *PALB2* genes can restore the frameshift mutations that often lead to non-functioning proteins. The secondary mutations usually occur away from the original mutation sites and create a chimeric version of the proteins. The acquisition of secondary mutations, including insertions or deletions that restore the gene expression in breast, ovarian and pancreatic cancers, is the main mechanism of PARPi and platinum-derived agent resistance [51,52]. Additionally, copy number gain and/or upregulation of the remaining functional allele in the *BRCA* gene has been shown in tumours with somatic loss of *BRCA* [53]. Re-expression of BRCA1 protein by de-methylation was demonstrated to occur at the time of recurrence in a patient with PARPi-resistant epithelial ovarian cancer [54]. The expression of the hypomorphic BRCA1 splice variant was demonstrated in preclinical studies as one of the PARPi resistance mechanisms. Genetic alterations in the highly conserved RING domain of *BRCA1* confer increased stability, with retained function and PARPi resistance [55].

Reversion mutations have similarly been observed in HR-associated protein RAD51. Functional restoration of RAD51 through mutations help to mediate homologous sequence invasion in the HR process. This results in the disruption of the synthetic lethal effects of PARPi treatment [56]. Another DNA repair effector molecule Early Mitotic Inhibitor 1 (EMI1) is associated with RAD51 stabilization and accumulation. Downregulation of EMI1 restores HR and thus PARPi resistance; the mutation occurs with high prevalence in triple-negative breast cancer treated with olaparib [57].

### 4.2. Stabilization of the Replication Fork

Upon replication stress, cells arrest to allow time for repair and re-entry into the cell cycle or apoptosis. In the absence of BRCA1 and BRCA2, stabilization of the replication fork by a number of other protein effectors induces alternate mechanisms of DNA repair resulting in PARPi resistance [58]. One of the molecules is FANCD2, which is often expressed in BRCA1/2 mutant ovarian, breast and uterine cancer cells. It functions by suppressing MRE11-mediated fork collapse [47]. In addition, SLFN11 is another important protein that stabilizes the replication fork by prolonging the S phase upon replication fork stress. Cells ultimately undergo irreversible interruption of the replication fork and become susceptible to the effects of PARP inhibitors [59].

### 4.3. Drug Efflux Pumps

Intracellular availability of drugs is mediated by efflux pumps. *ABCB1* genes, also referred to as multidrug resistance (MDR1) genes, encode for p-glycoprotein efflux pumps that are responsible for PARPi resistance, especially in BRCA1-deficient breast cancer and ovarian cancer cell lines [49,60]. It has been demonstrated that the ABCB1 transporters are upregulated in non-naïve (e.g., previously treated with chemotherapy) tumours as a result of chromosomal translocations, intergenic deletions and 5′ region mutations that occur upon paclitaxel treatment; hence, patients treated with chemotherapy prior to initiating PARPi therapy may experience increased risk for resistance to PARPi [61]. In both olaparib- and paclitaxel-treated ovarian cancer cells, verapamil and elacridar MDR1 inhibitors can reverse ABCB1-mediated drug resistance, but the clinical trial outcome is poor. These important findings suggest that PARPi drugs are not a substrate for ABCB1-mediated cellular efflux.

### 4.4. Downregulation of NHEJ

Mutations in the components of the NHEJ pathway can also lead to the reactivation of HR in BRCA1/2 mutant cells. 53BP1 is antagonized by BRCA1 to favour HR instead of NHEJ [62]. 53BP1 promotes NHEJ by limiting the DNA end resection, which is required for HR to occur. 53BP1 together with other effectors (RIF1, REV7, SHLD1, SHLD2, SHLD3) inhibits resection; the loss of any of the factors in this complex has been associated with PARPi resistance in BRCA1-deficient cells, but interestingly not in BRCA2-deficient cells [44,45,63,64]. Similar to the inactivation of 53BP1, the loss of another DNA end resection protein, Dynein Light Chain 1 (DYNLL1), has been associated with PARPi resistance [65].

### 4.5. PARP1 and PARG Mutations

Mutations in the PARP1 molecule itself have been linked with the development of PARPi resistance, particularly by decreasing PARP trapping on damaged DNA segments [48]. Mutations were frequently seen involving codon K119 and S120 and the surrounding region, impacting the ability of PARP1 to bind sites of DNA damage [48]. Specifically, there are specific types of *BRCA1* mutations that can tolerate mutations in PARP better than others. Mutations in exon 11 of *BRCA1*, as compared with frameshift mutated *BCRA1* genes, pose the risk of developing PARPi resistance with concomitant *PARP1* mutations. Cells harbouring *BRCA1* frameshift mutations will not be viable as the first place to be resistant to PARPi [48]. PARG plays a critical role in preventing and counteracting the effects of PARP1 by inducing the degradation of nuclear PAR. It was demonstrated that PARG depletion restored PARP1 signalling and led to PARPi resistance in HR-deficient tumours [66]. The restoration of PARP1 catalytic activity prevented uncontrolled replication fork progression and resumed the recruitment of downstream DNA repair factors, hence leading to PARPi resistance.

These resistance mechanisms are illustrated in Figure 1 and Figure 2 below.

## 5. Overcoming PARP Inhibitor Resistance through Combination Strategies

The utility of combinatorial strategies has the potential to overcome de novo and acquired PARPi resistance. Several agents have shown promise in this area either by amplifying the antitumour effects of PARPi or by targeting alternate pathways of HR repair. Here we discuss several combinatorial strategies of PARPi combination with chemotherapy, anti-angiogenesis agents, other DNA damage repair agents, the cell cycle and with immune checkpoint inhibitors.

### 5.1. PARPi and Chemotherapy

PARPi and chemotherapy can work synergistically, either as PARPi acting as a chemosensitizer, or chemotherapy eradicating PARPi-resistant cells. However, the benefits of this combination strategy are greatly offset by the overlapping toxicity, specifically myelosuppression, as observed in the Velia trial. In this study, patients were randomised 1:1:1 to receive combined PARPi, veliparib with induction chemotherapy (carboplatin and paclitaxel) followed by veliparib maintenance compared to induction chemotherapy with veliparib and placebo maintenance or induction chemotherapy without veliparib and placebo maintenance in patients with newly diagnosed ovarian cancer. The study demonstrated an improvement in PFS in the patients who received veliparib throughout (with induction chemotherapy and as maintenance) with the benefit not isolated to just the *BRCA*1/2 mutation carriers who had a median of 34.7 months vs. 22 months (hazard ratio for progression or death, 0.44; 95% confidence interval (CI), 0.28 to 0.68; *p* < 0.001), and also in the intention to treat population as well with PFS of 23.5 months compared to 17.3 months (hazard ratio, 0.68; 95% CI, 0.56 to 0.83; *p* < 0.001 [67]. However, the combination with veliparib led to a higher incidence of anaemia, thrombocytopenia, nausea and fatigue, making it a less favourable approach [67,68,69,70,71].

### 5.2. PARPi and ATRi

ATR complements PARP by stabilizing the replication fork during replication stress. It also activates S and G2-M checkpoints to allow DNA repair [72]. By inhibiting ATR, the replication fork destabilises leading to double-strand breaks and eventual cell death. ATR also overcomes SLFN11 inactivation, an expression that induces an irreversible and lethal replication inhibition independent of the ATR-mediated S phase checkpoint [73,74].

The CAPRI trial, which evaluated ATRi, ceralasertib with olaparib demonstrated a tolerable safety profile; however, the activity was modest with no objective responses demonstrated and the median PFS observed was 4.2 months overall (90% CI: 3.5–8.2) and 8.2 months (3.6 months–not determined) for patients with *BRCA1* mutations [75]. Though promising, larger randomised trials need to be conducted to attain more conclusive results.

### 5.3. PARPi and Immune Checkpoint Inhibitors

The rationale for this combination stems from the assumption that HRD cancers have a higher level of genomic instability resulting in elevated neo-antigen loads stimulating an increased antitumour immune response [76]. Treatment with PARPi synergises with immune checkpoint inhibitors to enhance immune surveillance in the body by promoting cross-presentation, modifying immune microenvironments and increasing PD-L1 expression [77,78].

Preliminary results from early phase basket studies demonstrate a tolerable safety profile for the combination of PARPi and immune checkpoint inhibitors with varying antitumour activity across different tumour types [79,80]. In the TOPACIO study, which explored the combination therapy of PARPi and pembrolizumab, patients with advanced or metastatic unselected TNBC demonstrated an ORR of 21% (90% CI, 12–33%) with a DCR of 49% [81], while the recurrent ovarian cancer cohort demonstrated an ORR of 18% (90% CI = 11–29%) and a DCR of 65% [82]. The MEDIOLA study evaluated the combination of olaparib and durvalumab (PD-L1 inhibitor) in patients across four tumour types. Among breast cancer patients, the median durations of response (DOR) of 9.2 months and median PFS of 8.2 months observed with durvalumab and olaparib [83] were of similar magnitude to the median (DOR) of 6.4 months and median PFS of 7.0 months observed in OlympiAD with olaparib monotherapy [84]. Within the *BRCA1/2* mutant ovarian cancer cohort, interim results suggest an ORR of 71.9% (95% CI = 53–86) [85]. Several larger randomised phase 3 trials with a combination of PARPi and ICI are underway, and the future translational work performed alongside these trials may allow for the better selection of patients based on a predictive biomarker of relevance.

### 5.4. PARPI and Anti-Angiogenesis Agents

The vascular endothelial growth factor (VEGF) protein promotes angiogenesis and increased vascular remodelling in response to hypoxic conditions. The rationale for the combination stems from the PARP1 pathway regulating gene expression associated with angiogenesis through hypoxia-inducible factors leading to VEGF-A upregulation [86]. The induction of hypoxia with an anti-angiogenic agent alters HR repair including the downregulation of BRCA1/2 and RAD51 [44,45], which has the potential to sensitise to PARPi treatment [87,88].

Several studies have evaluated the combination of PARPi and anti-angiogenic agents, mainly in ovarian cancer. In a randomized phase 2 trial comparing PARPi, olaparib against the combination, olaparib/cediranib (anti-angiogenic agent), in platinum-sensitive recurrent high-grade serous ovarian cancer (HGSC) patients (NCT01116648), an overall PFS of 17.7 months in the combination group and 9.0 months in the olaparib monotherapy group was observed [89]. Further exploratory analysis by BRCA1/2 status demonstrated that the combination was also active in BRCA1/2 wild type cohorts. A greater benefit was observed in the BRCA1/2 wild type cohort from the addition of cediranib, increasing the median PFS from 5.7 months with olaparib monotherapy to 16.5 months (*p* = 0.008) with combination therapy, while the BRCA1/2 mutant cohort showed a smaller improved median PFS from 16.5 months to 19.4 months (*p* = 0.16), respectively, in monotherapy compared with the combination [89]. In a separate phase II study of combination PARPi and an anti-angiogenic agent, the combination of PARPi, niraparib and an anti-angiogenic agent, bevacizumab, as treatment (rather than maintenance therapy) significantly improved the response rate compared to niraparib alone in the ANANOVA2 trial (60% vs. 27%), regardless of HR deficiency status [90], reaffirming the potential utility of this combination strategy in patients beyond *BRCA1/2* mutation

The combination of cediranib plus olaparib has been shown to potentially overcome resistance to PARPi in a small group of patients. This combination was evaluated in the EVOLVE study following progression on prior PARPi. Clinical benefit was observed in a few patients [91]. However, when paired biopsies obtained at the time of PARPi progression were analysed, several genomic alterations were observed including 19% with HR gene reversion mutations (BRCA1, BRCA2 or RAD51B), 16% with CCNE1 amplification, 15% with ABCB1 upregulation, and 7% with SLFN11 downregulation. Importantly, translational findings suggests that HR gene reversion mutations, or upregulated ABCB1 at trial baseline, resulted in patients with poorer outcomes, and they are unlikely to benefit from the cediranib/olaparib combination [91]. These findings suggest different types of acquired PARPi resistance require different treatment approaches and combination strategies.

These studies suggest that, as a form of treatment, PARPi combined with anti-angiogenic agents demonstrates efficacy based on objective response rates and PFS. Further translational studies will help highlight which patients will benefit most from this combination.

### 5.5. PARPI and PI3K

Another commonly dysregulated pathway in cancers is the phosphatidylinositol-4,5-biphosphate 3-kinase (PI3K)/AKT/mTOR pathway. Preclinical studies demonstrated a potential synergism between PARPi and PI3K pathway inhibitors by downregulating *BRCA1/2* and inducing an impaired DNA damage response and deficient homologous recombination repair [92,93].

Most of these combinations are still in early phase trials with promising results. A phase 1b clinical trial with alpelisib, a specific PI3K alpha inhibitor, in combination with olaparib in previously pretreated (median three lines) advanced triple-negative breast cancers had a median duration of response of 7.4 months, with 59% achieving disease control [94]. In the same study of 34 patients with recurrent epithelial ovarian cancer, alpelisib in combination with olaparib demonstrated a partial response of 36% with 50% achieving stable disease [95]. The ComPAKT trial evaluated the combination of capivasertib, an AKT inhibitor, with olaparib in advanced solid malignancies, and demonstrated that 44.6% of evaluable patients achieved a clinical benefit of either a shrinkage of the tumour by at least 30% or stable disease of 4 months or longer. Treatment benefits were seen in patients regardless of *BRCA1/2* mutation status or PI3K/AKT pathway alterations [96]. In another study that explored the same combination in endometrial, ovarian and triple-negative breast cancers, a similar clinical benefit rate of 41% was demonstrated with mTOR pathway activation and high receptor tyrosine kinase activity levels associated with resistance to the combination therapy [97]. Further studies will need to be conducted to assess the efficacy of PARPI and PI3K inhibitor combination.

### 5.6. Inhibition of Polymerase Theta (POLQ) Synergizes with PARPi in Eliminating HR-Deficient Tumours

Alternative non-homologous end-joining (alt-NHEJ) or microhomology-mediated end-joining (MMEJ) has been described as the non-canonical double-strand break (DSB) repair pathways [98]. MMEJ is an error prone DSB repair pathway and polymerase theta (POLQ) is identified as the key player in the mechanism [99]. A level of POLQ was observed to be overexpressed in homologous-recombinant-deficient (HRD) tumours and was associated with a poorer outcome [100,101,102]. Hence, inhibition of POLQ exhibited synthetic lethality in HRD tumours [103]. In a study using human and murine HR-deficient pancreatic ductal adenocarcinoma (PDAC) in vitro models, POLQ knockdown was shown to elicit synthetic lethality and induced DNA damage in HR-deficient tumour cells while simultaneously stimulating an immune response by activating the STING signalling pathway [104]. Another study also identified the antibiotic novobiocin (NVB) that works as a specific POLQ inhibitor that selectively kills BRCA-deficient breast and ovarian tumour cells in vitro and in vivo. In PARPi-resistant tumour cells, NVB mediated cell death by the accumulation of single-strand DNA intermediates and non-functional RAD51 foci. Hence, it was demonstrated that NVB may be useful alone or work synergistically with PARPi in treating HR-deficient tumours [105].

## 6. Cost-Effectiveness

Though clinical efficacy of PARPI has demonstrated its effectiveness and efficacy in treating ovarian and breast cancer, the drugs, however, are costly. A systematic review conducted by Haiying et al. reviewed 25 papers (17 on ovarian cancers, 3 on pancreatic cancer, 3 on prostate cancer and 2 on breast cancer) on the cost-effectiveness of the use of PARPI. PARPIs that were involved in the study included olaparib, rucaparib, talazoparin and niraparib [106]. The majority of the studies demonstrated the cost-effectiveness of olaparib in the setting of newly diagnosed patients after first-line platinum-based therapy; however, cost-effectiveness was reduced in patients with platinum-sensitive recurrences. The cost-effectiveness was less evident with other PARPIs in ovarian cancer and other cancer types. Factors that influence the cost-effectiveness are genetic testing, health utility and discount rate, survival time and the initial cost of the PARPi inhibitor.

Wolford et al. also demonstrated this when comparing the cost-effectiveness of niraparib, rucaparib and olaparib vs. non-platinum-based chemotherapy in platinum-resistant, recurrent ovarian cancer [107]. Chemotherapy was the most effective at USD 6412/PFS-month), followed by bevacizumab-containing regimens (USD 12,187/PFS-month), and then the PARPIs (olaparib at USD 16,327/PFS-month, rucaparib at USD 16,637/PFS-month and niraparib at USD 18,970/PFS-month). The incremental cost-effectiveness ratios (ICERS) for PARPI were 3–3.5× greater than non-platinum-based chemo.

A study by Rafael Gonzalez et al. proposed a biomarker-directed use of PARPi maintenance therapy for newly diagnosed advanced stage ovarian cancer to increase its cost-effectiveness. The study used Markov decision models that simulate study designs by PAOLA-1, PRIMA and VELIA trials while evaluating overall cost, ICERs, in US dollars per quality-adjusted progression-free life year (QA-PFY) gained. This demonstrated that although PARPi for all provided a greater PFS benefit, it was more costly with mean costs per patient of USD 166,269, USD 286,715 and USD 366,506 for the PRIMA, VELIA and PAOLA-1 models, respectively. This is in comparison to the biomarker-directed strategy with mean costs per patient of USD 98,188, USD 167,334 and USD 260,671 for the PRIMA, VELIA and PAOLA-1 models, respectively. The ICERs were USD 593,250/QA-PFY for PRIMA, USD 1,512,495/QA-PFY for VELIA and USD 3,347,915/QA-PFY for PAOLA-1 [108].

## 7. Conclusions

In conclusion, PARPi therapy has helped transform the treatment of certain cancers and provided a promising strategy with tolerable side effects, with potentially new PARPI in the works that may mitigate these side effects such as AZD2461 [109]. Multiple mechanisms of resistance to these agents have been identified in the preclinical setting with ongoing confirmation of the clinical relevance in translational studies. Combination treatment offers a potential strategy to overcome PARPi resistance. However, an improved understanding of the underlying mechanisms of resistance to PARPi and the synergistic mechanism of response of combinatorial strategies is urgently required. Further studies with parallel translational research including collection of tumour tissue +/− ctDNA are required to better define predictive biomarkers and facilitate patient stratification for these combination therapies. 

## Figures and Tables

**Figure 1 cancers-15-02357-f001:**
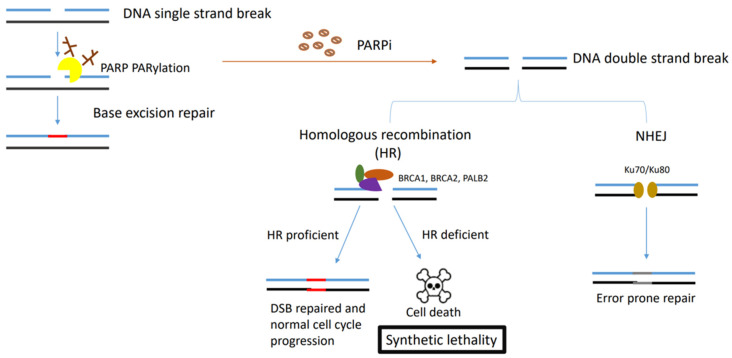
Figure describing the synthetic lethality interaction between PARPs and *BRCA1/2*. PARP binds to the single-strand DNA break sites, and results in the PARylation of target proteins and recruitment of the DNA damage repair effectors. In HR-deficient tumour cells (BRCA1-, BRCA2- or PALB2-deficient cells) treated with PARPi, NHEJ is the only pathway initiated in double-strand DNA repair, which leads to accumulation of genome instability and cell death for the low fidelity.

**Figure 2 cancers-15-02357-f002:**
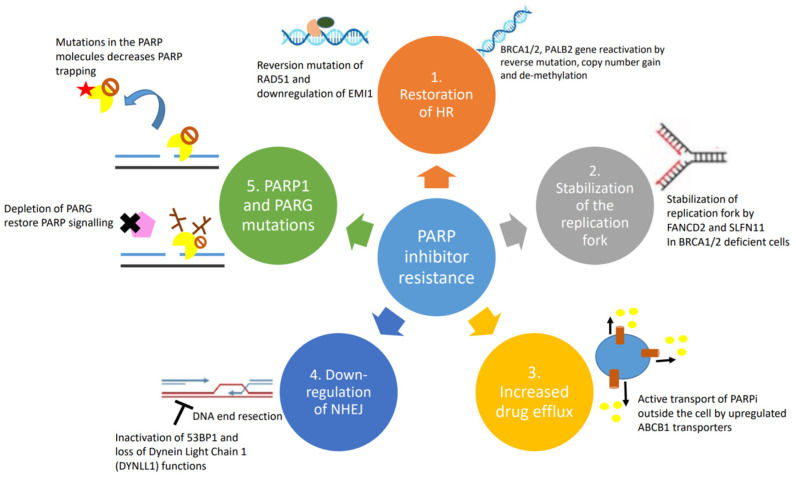
Causes and mechanisms of PARP inhibitor resistance: (1) restoration of homologous recombination; (2) stabilization of replication fork; (3) upregulation of drug efflux pumps; (4) downregulation of NHEJ; and (5) PARP1 and PARG mutations.

**Table 1 cancers-15-02357-t001:** Key trials demonstrating use of PARPI inhibitors in cancer treatment.

Tumour Type	Drug	Population	Clinical Setting	Significant Results	Study Reference
Breast cancer	Veliparib	- Triple-negative breast cancer. - BrighTNess Trial	- Neoadjuvant treatment of stage II-III triple-negative breast cancer	- Veliparib + carboplatin + paclitaxel arm (*n* = 316): CR ^1^ 53% - Paclitaxel + carboplatin arm (*n* = 160): CR 49% - Paclitaxel alone arm (*n* = 158): 58%	[17]
Breast cancer	Olaparib	- BRCA1 or BCRA2 mutated breast cancer. - OlympiA trial	- Adjuvant therapy for 1 year for patients that had received local treatment, and neoadjuvant/adjuvant chemo	- Oliparib arm (*n* = 918): 3-year invasive DFS ^2^ 85.9%, 3-year distant DFS 87.5%, deaths *n* = 59 - Placebo arm (*n* = 918): 3-year invasive DFS 77.1%, 3-year distant DFS 80.4%, death *n* = 86	[18]
Breast cancer	Talazoparib	- Advanced breast cancer and a germline BRCA1/2 mutation - EMBRACA trial	- Advanced/metastatic setting - Either locally advanced breast cancer that had not been amenable to curative therapy or metastatic breast cancer	- Talazoparib arm (*n* = 287): median PFS ^3^ 8.6 months, objective response rate 62.6% - Standard treatment arm *(n* = 144): median PFS 5.6 months, objective response rate 27.2% - Interim hazard ratio for death: 0.76	[16]
Breast cancer	Olaparib	- Advanced breast cancer - OLYMPIAD study	- Metastatic breast cancer who had received 2 lines or less of chemotherapy	- Olaparib arm (*n* = 205): median OS 19.3 months, low rates of treatment discontinuation - Standard treatment arm (*n* = 97): median OS 17.1 months	[15]
Ovarian cancer	Niraparib	- Advance ovarian cancer - PRIMA trial	- Maintenance treatment for ovarian cancer irrespective of BRCA carrier status - 50.9% had homologous recombination deficiency	- Niraparib arm (*n* = 484): PFS 13.8 months, overall survival 84% - Placebo arm (*n* = 175): PFS 8.2 months, overall survival 77% - PFS was superior in patients that had homologous recombination deficiency (21.9 months vs. 10.4 months)	[19]
Ovarian Cancer	Olaparib	- Ovarian Cancer - SOLO1 trial	- Newly diagnosed advance ovarian cancer with maintenance Olaparib	- Oliparib arm (*n* = 260): median duration of treatment 24.6 months, median PFS 56 months - Placebo arm (*n* = 131): median duration of treatment 13.9 months, median PFS 13.8 months	[20]
Ovarian Cancer	Olaparib	- Advance high-grade ovarian cancer - POALA trial	- Newly diagnosed advanced, high-grade ovarian cancer with maintenance Olaparib + bevacizumab	- Olaparib + bevacizumab arm (*n* = 537): median PFS 22.1 months - Placebo + bevacizumab arm (*n* = 269): median PFS 16.6 months - HR for PD or death 0.33 in patients positive for HRD (median PFS, 37.2 vs. 17.7 months)	[21]
Ovarian cancer	Rucaparib	- Newly diagnosed stage III-IV high-grade ovarian cancer with R/0 resection responding to first-line platinum doublet - ATHENA-MONO	- Maintenance rucaparib for newly diagnosed stage III-IV high-grade ovarian cancer	- Rucaparib arm (*n* = 427): median PFS 28.7 months, 20.2 months in the intent to treat population, 12.1 months in HRD negative population - Placebo arm (*n* = 111): median PFS 11.3 months, 9.2 months in intention to treat population, 9.1 months in HRD negative population	[22]
Ovarian Cancer	Olaparib	- Platinum-sensitive relapsed ovarian cancer and a germline BRCA1/2 mutation - SOLO trial	- Olaparib vs. non-platinum chemotherapy BRCA-mutated platinum-sensitive relapsed ovarian cancer who had received at least 2 prior lines of platinum-based chemotherapy.	- Olaparib arm (*n* = 178): overall ORR ^4^ 72.2%, ORR 84.6% in subgroup that received 2 prior lines of treatment - Chemotherapy arm (*n* = 88): overall ORR 51.4%, ORR 61.5% in subgroup that received 2 prior lines of treatment	[23]
Ovarian Cancer	Olaparib	- Platinum-sensitive relapsed ovarian cancer and a germline BRCA1/2 mutation - SOLO2 trial	- Olaparib maintenance for all patients with platinum-sensitive, relapsed ovarian cancer and a BRCA1/2 mutation.	- Olaparib arm (*n* = 196): overall PFS 19.1 months [95% CI 16.3–25.7] - Placebo arm (*n* = 99): overall PFS (5.5 months [5.2–5.8]; hazard ratio (HR) 0.30 [95% CI 0.22–0.41], *p* < 0.0001	[24]
Ovarian Cancer	Olaparib	- Platinum-sensitive, relapsed, high-grade serous ovarian cancer who had received two or more platinum-based regimens and had had a partial or complete response. - Study 19 trial	- Olaparib as maintenance treatment for patients with platinum-sensitive, relapsed ovarian cancer	- Overall PFS olaparib arm (*n* = 136) 8.4 months vs. placebo arm (*n* = 129): 4.8 months [5.2–5.8]; HR, 0.35; 95% confidence interval (CI), 0.25 to 0.49	[25]
Ovarian Cancer	Niraparib	- Platinum-sensitive, recurrent ovarian cancer - NOVA	- Niraparib for maintenance treatment for patients with platinum-sensitive, recurrent ovarian cancer.	- 203 were in the germline BRCA cohort, the overall PFS *n* = 138; niraparib arm 21.0 months vs. *n* = 65 placebo arm 5.5 months (HR, 0.27; 95% (CI), 0.17 to 0.41 - 350 non-germline BRCA cohort *n* = 234; niraparib arm 12.9 months vs. *n* = 116 placebo arm 3.8 months - (HR, 0.45; 95% CI, 0.34 to 0.61; *p* < 0.001) - In patients with homologous recombination deficiency in the non-germline BRCA - 12.9 months for niraparib arm vs. 3.8 months placebo arm (HR, 0.38; 95% CI, 0.24 to 0.59)	[26]
Ovarian Cancer	Rucaparib	- BRCA mutant or BRCA wild type and loss of homozygosity (LOH) high platinum-sensitive ovarian carcinomas - ARIEL 2	- Rucaparib for maintenance treatment for patients with platinum-sensitive, recurrent ovarian cancer.	- Median PFS 12.8 months (95% CI 9.0–14.7) in the BRCA mutant subgroup *n* = 40, 5.7 months (5.3–7.6) in the LOH high subgroup *n* = 82 and 5.2 months (3.6–5.5) in the LOH low *n* = 70 subgroup. PFS was significantly longer in the BRCA mutant (hazard ratio 0.27, 95% CI 0.16–0.44, *p* < 0.0001) and LOH high (0.62, 0.42–0.90, *p* = 0.011)	[27]
Ovarian Cancer	Rucaparib	- Recurrent ovarian carcinoma after response to platinum therapy - ARIEL3	- Rucaparib maintenance treatment for recurrent ovarian carcinoma after response to platinum therapy	- Rucaparib arm (*n* = 375): median PFS 16.6 month, 13.6 month in HRD group - Placebo arm (*n* = 189); median PFS 5.4 months, 5.4 months in HRD group	[28]
Prostate Cancer	Olaparib	- Prostate Cancer - PROfound trial	- Metastatic castrate resistant prostate cancer with alteration in genes that affect homologous recombination repair	- Cohort A (BCRA 1, BRCA2, or ATM) vs. control: PFS 7.4 months vs. 3.6 months, median OS ^5^ 18.5 months with olaparib vs. 15.1 months	[29]
Pancreatic Cancer	Olaparib	- Pancreatic Cancer - POLO trial	- Metastatic BCRA mutated pancreatic cancer	- Olaparib arm (*n* = 92): median PFS 7.4 months - Placebo arm (*n* = 62): median PFS 3.8 months - Interim overall survival at data maturity 46%: no difference n survival - No difference in health-related quality of life	[30]

Abbreviations in table: (^1^) CR: complete response. (^2^) DFS: disease-free survival. (^3^) PFS: progression-free survival. (^4^) ORR: objective response rate. (^5^) OS: overall survival.

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
