# Peer review of "PARP Inhibitors in Breast and Ovarian Cancer"

_cancers, 2023, doi:10.3390/cancers15082357_

Round 1

Reviewer 1 Report

The authors should mention the new clinical trials on other cancers in Clinical trials.gov. (400 trials) Also PARP-1 has been shown to be involved in homologous recombination and in binding directly to double strand breaks.

The cost versus benefit of the use of PARP inhibitors in cancer treatment should be discussed. The new PARP-1 inhibitor developed by Astra Zeneca

should be mentioned since some of the side effects are removed compared to other PARP inhibitors.

Author Response

We would like to thank you for your time in reading our review and for invaluable comments. We have revised the manuscript as suggested below:

  • We have made mentioned azd2461 in conclusions on the progress of parpi with more tolerable side effect
  •  Added on a section regarding cost effectiveness of PARPI inhibitors as below:
    •      Though clinical efficacy of PARPI has been demonstrated its effectiveness and efficacy in treating ovarian and breast cancer, the drugs however are costly and often patients will have problems affording it.

      A systematic review done by Haiying Et Al reviewed 25 papers (17 on ovarian cancers, 3 on pancreatic cancer, 3 on prostate cancer, and 2 on breast cancer) on the cost effectiveness of the use of PARPI. PARPIs that have been involved in the study included olaparib, rucaparib, talazoparin and niraparib (106). The majority of the studies have demonstrated cost effectiveness of oliparib in the setting of newly diagnosed patients after first line platinum based therapy, however the cost effectiveness is reduced in patients with platinum sensitive recurrences. The cost-effectiveness was less evident with other PARPIs in ovarian cancer and other cancer types. Factors that influence the cost effectiveness are genetic testing, health utility and discount rate, survival time, and the initial cost of the PARPi inhibitor. 

      Wolford Et Al has also demonstrated this when comparing the cost effectiveness of niraparib, rucaparib and olaparib vs non- platinum based chemotherapy in platinum resistant, recurrent ovarian cancer (107). Chemotherapy was the most effective at $6,412/PFS-month), followed by bevacizumab containing regimens ($12,187/PFS-month), then the PARPIs (Olaparib at $16,327/PFS-month, Rucaparib at $16,637/PFS-month, and niraparib at $18,970/PFS-month). The incremental cost-effectiveness ratios (ICERS) for PARPI was 3-3.5x greater than non-platinum based chemo.

      A study by Rafael Gonzalez et al proposes a biomarker directed use of PARPI maintainence therapy for newly diagnosed advanced stage ovarian cancer to increase its cost effectiveness. The study used Markov decision models that simulate study designs by PAOLA-1, PRIMA and VELIA trials while evaluating overall cost, ICERs in US dollar per quality adjusted progression free life year (QA-PFY) gained. It has demonstrated though PARIPI for all provided greater PFS benefit, it was more costly with a mean cost per patient of $166,269, $286,715, and S366,506 for the PRIMA, VELIA and PAOLA-1 models respectively. This is in comparison to biomarker directed strategy having amean cost per patient of $98,188, $167,334, and $260,671 for the PRIMA, VELIA, and PAOLA-1 models. The ICERs were $593,250/QA-PFY for PRIMA, $1,512,495/QA-PFY for VELI, and $3,347,915/QA-PFY for PAOLA-1 (108).

Reviewer 2 Report

Wang and colleagues write a review on PARP inhibitors in breast and ovarian cancers. The review is well-written and structured to be of interest for readers. Authors focus on PARP inhibitors and the resistance to this therapy. New combinations to overcome PARP inhibitors resistance are described. I here below list a few minor issues that, hopefully, will help authors to finalize their manuscript before publication.

1. It lacks a section on DNA polymerase theta (POLQ) targeting in cancers (in BRCA cancers especially). Discussing recent work from Vincenzo Costanzo, Raphaël Ceccaldi, Alan D’Andrea and others. Here below is a list of papers that should be discussed in section 5.

PMID: 36976649 ; PMID: 34179826 ; PMID: 25642963 ; PMID: 36400008

2. Page 5. The drugs enzalutamide and abiratenone when discussing ref (31) are not described to non-specialists.

3. Table 1. This table should be re-sized to fit a smaller table format that can integrate on a single page.

Author Response

We would like to thank you for your time in reading our review and for invaluable comments. We have revised the manuscript as suggested below

  • We have added a section on POLQ to section 5
  • We have added that abiraterone is a androgen biosynthesis inhibitor, that inhibits 17α-hydroxylase/C17,20-lyase (CYP17).
  • Table has been shrunk down to 2 pages with size 6, could not reduce the size of table further

Reviewer 3 Report

The manuscript of the article is devoted to the topical issue of Poly(ADP-ribose) polymerase (PARP) inhibitors in oncology. The material fully corresponds to the goals and direction of the journal. I studied this review with interest and found a lot of useful things for myself. I hope that this manuscript will become a published article as soon as possible.

Author Response

We would like to thank you for your time in reading our review and for your kind comments.